# Overview of Lightning Trend and Recent Lightning Variability over Sri Lanka

**Vindhya Kalapuge** [1], **Dilaj Maduranga** [2], **Niranga Alahacoon** [3], **Mahesh Edirisinghe** [1,*],
**Rushan Abeygunawardana** [4] and **Manjula Ranagalage** [5]

1. Department of Physics, University of Colombo, Colombo 00300, Sri Lanka
2. Department of Physics, The Open University of Sri Lanka, Nawala, Nugegoda 10250, Sri Lanka
3. International Water Management Institute (IWMI), 127, Sunil Mawatha, Pelawatte, Batteramulla, Colombo 10120, Sri Lanka
4. Department of Statistics, University of Colombo, Colombo 00300, Sri Lanka
5. Department of Environmental Management, Faculty of Social Sciences and Humanities, Rajarata University of Sri Lanka, Mihintale 50300, Sri Lanka
* Correspondence: mahesh@phys.cmb.ac.lk

**Abstract:** The study was conducted to analyze spatial and temporal variations of lightning activity over Sri Lanka and the surrounding coastal belt region bounded by 5.75–10.00 N and 79.50–89.00 E. Flash data collected by the Lightning Imaging Sensor (LIS) on NASA's Tropical Rainfall Measuring Mission (TRMM) satellite from 1998 to 2014 and the Lightning Imaging Sensor placed on the International Space Station (ISS) from 2018 to 2021 were used for the study. The Mann-Kendall test and Sen's slope estimator were applied to annual and seasonal lightning data from 1998 to 2014 to identify the trends in the TRMM dataset. A positive slope of 0.23 was obtained for annual flash densities, while a slope of 0.956 was obtained for First Inter-Monsoon (FIM) seasonal data. Considering the ISS data, the annual variation of lightning activity in 2020 displays the lowest flash density, whereas the highest is represented in 2019 with a value of 10.48 flashes $km^{-2}$ $year^{-1}$. The highest mean flash density is observed in Colombo in 2019 at a value of 34.85 flashes $km^{-2}$ $year^{-1}$. Overall, April displayed the highest annual flash distribution from 2018 to 2021, whereas the second peak was mostly viewed around September and November. All districts have displayed a significant amount of lightning during April for the period 2018 to 2021. FIM displayed the highest lightning distribution over the country. When considering the seasonal variation, districts belonging to the wet zone and intermediate zone displayed most flashes during the FIM.

**Keywords:** lightning; lightning hazards; lightning variability; lightning safety; lightning flash density; lightning trend; Mann-Kendall test; Sen's slope estimator; LIS data

## 1. Introduction

Lightning is a natural phenomenon that occurs often on earth, and a large thunderstorm can produce more than a hundred lightning flashes within a very small time duration. Thunderstorms, together with the associated lightning and gusts, are picturesque but potentially devastating meteorological occurrences. Those events can be very hazardous to living beings and properties because current pulses that occur have the potential to cause damage. Therefore, comprehensive knowledge about the spatial and temporal distribution of lightning activity and variation over a country in recent years is important in developing appropriate protective measures.

Nonetheless, reducing lightning risks and developing appropriate lightning protection systems in a given location requires initially conducting a thorough analysis of the regional diversity in lightning activity [1–3]. Lightning detection systems are useful to study the distribution and variation of lightning activity monthly, annually, and seasonally over the country. Globally, Orville analyzed the distribution of ground lightning flashes over

the contiguous United States using the cloud-to-ground (CG) lightning flash data of the National Lightning Detection Network (NLDN) from 1995 to 1997 [4]. Lightning is one of the most frequently-occurring phenomena around the globe. Some 20% of the lightning reaches the ground, causing an annual number of approximately 24,000 fatalities and many injuries [5]. Lightning can also cause damage to buildings and structures. To identify the risks of lightning and conduct assessments on them, it is essential to continue studies on risk assessment and build lightning risk models. A lightning risk assessment program can be developed by considering parameters such as lightning data, facility data and topographical data, and performing evaluations on lightning hazard levels and risks [6]. The study conducted by Hu focused on performing a lightning risk assessment in a residential area by focusing on the lightning events and the ground sensitivity of the area [7]. Building lightning protection systems while analyzing the risk assessment models is also important to mitigate lightning hazards. Therefore, it is crucial to understand the patterns of electrical discharges during lightning strikes, as they have a great impact on designing lightning warning and protection systems. In his study, Cummins explained the use of NLDN data to send alerts to electric utilities during lightning storms and evaluate the efficiency of the lightning protection systems [8]. In the Sri Lankan context, there are no lightning detection systems. Therefore, lightning satellite data from the Tropical Rainfall Measuring Mission (TRMM) and International Space Station (ISS) are important to analyze the trend of lightning activity over Sri Lanka. Furthermore, the information in these studies is significant to a variety of entities, including energy production organizations, construction firms, and local government agencies, as they prepare to launch important projects.

Lightning activity is not distributed evenly around the world; it varies due to longitude, latitude, altitude, soil content, topography, and, the vegetation of the region [9]. Variations of soil moisture caused due to soil content and vegetation have been suggested as one of the main reasons for the occurrence of CG lightning. However, more comprehensive studies need to be conducted to identify the significance of these parameters in relation to lightning incidents [10]. According to previous studies, about 70% of lightning activity occurs over land in the tropics [11]. Furthermore, the process of thunderstorm formation over the tropics is different from that of temperate regions in the world and it depends on the variations of Inter-Tropical Convergence Zone (ITCZ). Sri Lanka is a tropical country that is situated near the equator. As a result, the frequency with which lightning strikes occur in Sri Lanka is higher than in most other regions. Previous studies show that lightning activity over Sri Lanka has a clear incremental trend [12,13]. In this study, comprehensive statistical tools are utilized to analyze the trend in the variation of lightning activity, broken down by month, season, and year.

One of the most vital parts of climate research is identifying, quantifying, and forecasting trends together with their statistical and physical importance. Statistical trend estimation methods contemplate climate activities in a certain manner, thereby enabling the discovery of patterns or trends. The widely-used empirical approach in trend calculation is linear regression, and it has contributed significantly to climate analysis. The Mann-Kendall test is one of the powerful nonparametric approaches adopted to detect monotonic trends in environmental and hydrological data. Sen's estimator is another nonparametric estimator that measures linear incline. It is often used with the Mann-Kendall test to determine the linear tendency and its slope [14]. Utilizing the Mann-Kendall test and the Theil-Sen test, Hurun, et al. were able to emphasize a better outcome in their statistical analysis of Canadian lightning and thunderstorm data [15]. It is now common practice to use machine learning tools with standalone machine learning algorithms to make predictions on climatological data with an acceptable level of accuracy [16].

A space-based instrument, the Lightning Imaging Sensor (LIS) on the TRMM and ISS, is capable of detecting the location of a lightning event, the time, the rate of the lightning flashes, and their associated radiant energy, by detecting the optical pulses of the lightning [17–20]. According to the previous study, associated lightning energy is obtained by considering the time-integrated lightning energy density [21]. For this study, TRMM

lightning data for the period from 1998 to 2014 and ISS lightning data from 2018 to 2021 were utilized to explore lightning trends over Sri Lanka. The variation of recent lightning activity investigated lightning trends, and forecasted lightning flash density values in this study are critical information for designing lightning protection systems, planning long-term construction projects, and proposing lightning safety guidelines, amongst other things. Furthermore, Geographic Information System (GIS) software is important to get a better understanding about the spatial variation of the geo-referencing lightning satellite data and obtaining the basic statistical information on the lightning flash density of the country in this study.

## 2. Materials and Methods

### 2.1. Study Area

Sri Lanka is an island off the southern coast of India in the Indian Ocean, between 5.000 and 10.000 north latitude and between 79.000 and 82.000 east longitude. Its total area is about 65,000 square kilometers. According to the topography of the country, the central region consists of mountains and the rest is made up of wide lowlands and coastal belts. As shown in Figure 1, Sri Lanka is divided into four major climate zones based on long-term annual rainfall. Annual rainfall received by the wet zone is 2500 mm, while the dry zone receives rainfall between 1200 and 1900 mm due to the northeast monsoon. The lowest rainfall is received by the semi-arid zone, at between 800–1200 mm, while the intermediate zone receives rainfall somewhere between 1200 and 2500 mm per year [22,23]. The lightning flash density of the wet zone is twice that of the dry zone. The highest number of lightning occurrences are observed in the densely-populated western region of the country [20]. Lightning and thunderstorms occur mainly during four major monsoon seasons in Sri Lanka: First Inter-Monsoon (FIM—March to April), Southwest Monsoon (SWM—May to September), Second Inter-Monsoon (SIM—October to November), and Northeast Monsoon (NEM—December to February). The highest frequency of lightning over the country occurs during the First Inter-Monsoon season from March to April, whereas the least number of lightning flashes occur during the Northeast Monsoon season [24]. Studies conducted to identify the spatial distribution of lightning flashes have focused on the distribution across administrative sectors of the country. The administrative hierarchy of Sri Lanka is as follows: Provinces, Districts, Divisional Secretariat Divisions (DSD), and Grama-Niladari Divisions (GND). Districts such as Gampaha, Colombo, Ratnapura, and Kegalle are more prone to lightning activity compared to others [20,24]. These administrative districts belong to one or more climatic zones as denoted in Figure 1, and they have various topographies and soil textures. Districts such as Rathnapura, Badulla, Nuwara Eliya and Puttalam display collections of various vegetation. Therefore, for this study, the spatial analysis is conducted based on country level and district level, as they represent various altitudes, longitudes, latitudes, vegetation, soil types, rainfall, and topographies. Identifying lightning variations using administrative districts will also benefit the government in taking safety precautions against lightning hazards.

### 2.2. Lightning Data

The Tropical Rainfall Measuring Mission (TRMM) space mission, designed by NASA, has improved our understanding of the distribution and variation of precipitation within the tropical and subtropical regions bounded between 350 N and 350 S since 1997 [25]. A space-based instrument, the Lightning Imaging Sensor (LIS) on the TRMM is used to detect total lightning activity (Inter-Cloud, Intra-Cloud, Cloud-to-Air, and Cloud-to-Ground [CG] lightning) by covering a region of 600 km × 600 km on earth while detecting lightning activity with a spatial resolution between 3 km and 6 km [24]. This instrument measures the lightning amount, location, rate, and associated energy. The orbit of the TRMM is inclined by 350 and it had an altitude of (350 ± 6) km from 1998 to August 2001 and (403 ± 6) km from August 2001 to 2014. Even though the altitude was altered, for this analysis, the mean

observation duration for an individual storm was considered as 80 s by considering the count of orbits observing a selected grid box.

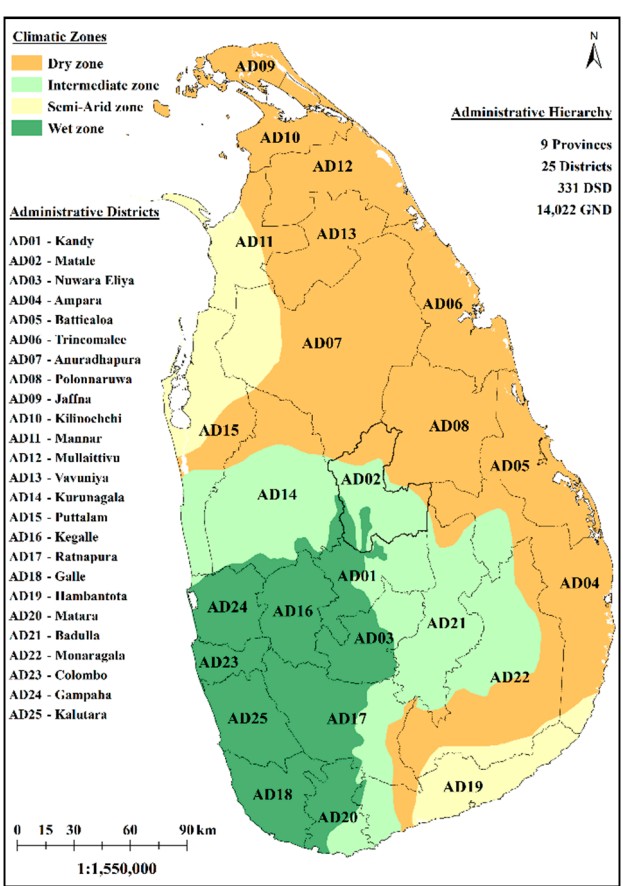

**Figure 1.** Study area indicating climatic zones, district boundaries, and administrative divisions of Sri Lanka.

The LIS on the TRMM covers Sri Lanka and its surroundings during a time window of 160 s per day (the LIS on the TRMM has an observational window duration of 80 s per single day and travels over Sri Lanka twice per day—hence, 160 s per day), while detecting total lightning discharges during day and night with an efficiency of 69% and 88%, respectively [24]. After the termination of the TRMM mission in 2015, another LIS instrument was placed on the International Space Station (ISS) in February 2017, which had a spatial extent of 55° N–55° S, a spatial resolution of 4–8 km, an altitude of 400–405 km and an inclination of 51.6°. The LIS on ISS has a 90 s window period for observations during one single storm and travels over Sri Lanka two times per day. It has a lightning detection efficiency of 64–75% during night-time and 51–65% during daytime [26]. For this study, total lightning data from 1998 to 2014 from the TRMM mission and total lightning data from 2018 to 2021 from the ISS were utilized, covering an area bounded by 5.75–10.00 N and 79.50–89.00 E.

*2.3. Trend Analysis*

Analyzing trends in climatological time series datasets which include uncertainties that have occurred due to many artificial inductions into the environment is a challenge. To date, parametric statistical methods such as moving averages and regression methods are most commonly used in detecting trends in temperature, rainfall, etc. A limitation of using these methods is that they require the data to have a normal distribution [27]. Non-parametric approaches such as the Mann-Kendall test can be applied to time series datasets having all types of distributions. These methods can be used to identify the significance of

the trend, increasing or decreasing with time. The magnitude of the slope of the trend can be calculated by applying the Theil-Sen approach, which has high resistance to outliers [15].

### 2.3.1. Mann-Kendall's Method (MK)

Mann-Kendall's method is a non-parametric statistical test used to identify positive or negative trends that could exist in a time series dataset. The null hypothesis ($H_0$) of this test states that all observations are independent, whereas the alternative hypothesis ($H_1$) states the presence of a monotonic trend, positive or negative, in the data [22,27–29]. The Mann-Kendall statistic value $S$ can be calculated by Equation (1):

$$S = \sum_{i=1}^{n-1} \sum_{j=i+1}^{n} sgn(x_j - x_i) \mid sgn(x_j - x_i) = \begin{cases} +1, & (x_j - x_i) > 0 \\ 0, & (x_j - x_i) = 0 \\ -1, & (x_j - x_i) < 0 \end{cases} \tag{1}$$

where $S$ is the Mann-Kendall statistic, *sgn* is the signum function, $x_j$ and $x_i$ are sequential data points and n is the number of data points. Obtaining a positive value of $S$ indicates that it has an upward trend, while a downward trend is denoted by a negative value of $S$.

Equation (2) can be used to calculate the variance of $S$ when the number of time-series records is greater than 10 ($N > 10$). This assumes that statistic $S$ is approximately normally distributed, making the average distribution of $S$ equal to zero.

$$Var(s) = \frac{n(n-1)(2n+5) - \sum_{i=1}^{m} t_i(t_i - 1)(2t_i + 5)}{18} \tag{2}$$

where $t$ denotes the number of tie groups specified in the sample and $t_i$ is the number of data points in the $i$th tie group. When $N > 10$, the standard average deviation $Z_c$ which indicates the availability of a significant trend in the dataset is calculated according to Equation (3).

$$Z_c = \begin{cases} \frac{S-1}{\sqrt{Var(S)}}, & S > 0 \\ 0, & S = 0 \\ \frac{S+1}{\sqrt{Var(S)}}, & S < 0 \end{cases} \tag{3}$$

$Z_c$ is normally distributed where a positive value for $Z_c$ indicates that it contains an increasing trend, whereas a negative value indicates a decreasing trend. The null hypothesis ($H_0$) can be rejected if $\mid Z_c \mid$ is greater than $Z_{1-\alpha/2}$ where $\alpha$ is the significance level for the test [30].

### 2.3.2. Sen's Slope Estimation (SSE)

Sen's slope estimator is used to identify the linear trend that exists in a time series dataset. Unlike linear regression, Sen's slope test is not affected by outliers and noisy data. Equation (4) can be used to calculate the trend of $N$ data sample pairs.

$$Q_i = \frac{(x_j - x_i)}{j - i} \mid i = 1, 2, 3, \ldots, N \tag{4}$$

$x_j$ and $x_i$ are data points at time $j$ and $i(j > i)$ respectively. $N = n(n-1)/2$ slope estimates are created if there are $n$ data points in the time series. Sen's slope is the median of $Q_i$ after arranging $Q_i$ values for each slope estimate in ascending order and calculated using Equation (5).

$$Q_{med} = \begin{cases} Q_{[\frac{N+1}{2}]}, & if\ N = odd \\ \frac{Q_{[\frac{N}{2}]} + Q_{[\frac{N+1}{2}]}}{2}, & if\ N = even \end{cases} \tag{5}$$

### 2.4. Methodology

Datasets collected from the ISS from 2018 to 2021 were used to generate district-wise and island-wise lightning flash density considering annual, seasonal (monsoonal), and monthly data. Seasonal and monthly patterns from 1998 to 2014 were identified using TRMM data and flash density raster maps were generated considering district and island-wise. A clear descriptive summary of the methodology followed is shown in Figure 2. Equation (6) has been used to calculate the average flash densities from 2018 to 2021 and Equation (7) has been utilized to calculate the average flash densities from 1998 to 2014. Both datasets were associated with 0.20 × 0.20 latitude and longitude grids to correspond with the previous studies [19]. According to [31,32], LIS-on-TRMM and LIS-on-ISS collection of data for different locations have varying view times between 0 to 80 s and 0 to 90 s respectively. In order to determine the maximum threat level due to lightning over the country, the highest values from the range (80 s and 90 s) were taken as the view times to calculate the lightning flash density, as shown in Equations (6) and (7).

$$\text{flash density}\left(\text{flashes}/\left(\text{km}^2 \cdot \text{year}\right)\right) = \frac{\sum \text{flashes}}{22\,\text{km} \times 22\,\text{km}} \times \frac{24 \times 3600}{2 \times 90} \times \frac{1}{\text{years}} \quad (6)$$

$$\text{flash density}\left(\text{flashes}/\left(\text{km}^2 \cdot \text{year}\right)\right) = \frac{\sum \text{flashes}}{22\,\text{km} \times 22\,\text{km}} \times \frac{24 \times 3600}{2 \times 80} \times \frac{1}{\text{years}} \quad (7)$$

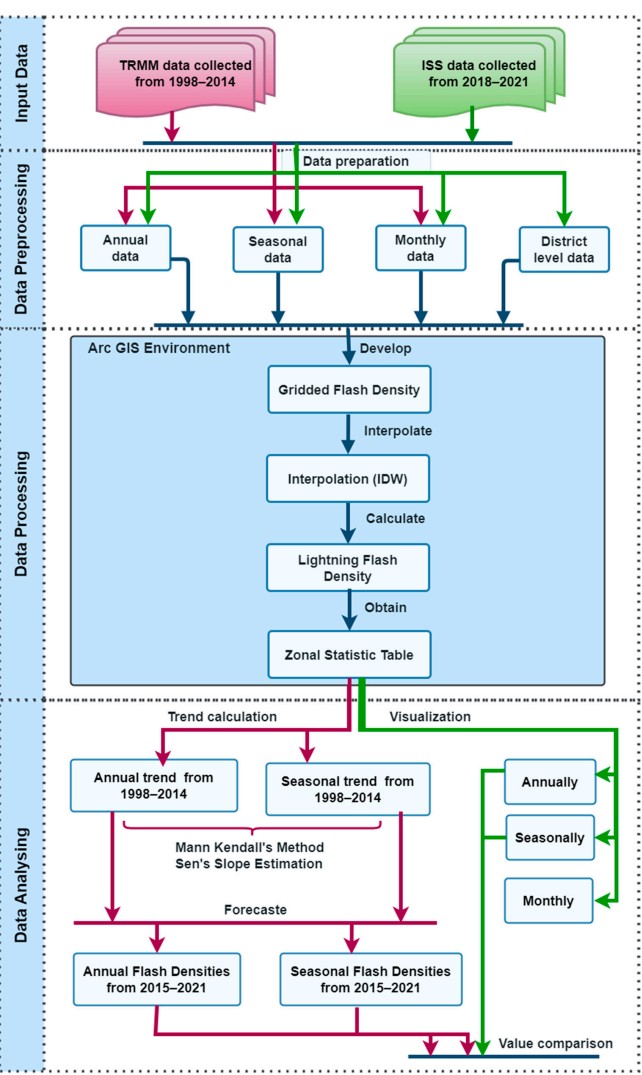

**Figure 2.** Methodology flow chart of the study.

Geographic Information System (GIS) was used to interpolate average lightning flash densities by using Inverse Distance Weighting (IDW) methodology. The zonal statistics tool in GIS was used to compute the minimum, maximum, average, and standard deviation of flash densities at the island level and district level. Python programming language was used to analyze the patterns in the datasets and used to apply the Mann-Kendall test to calculate the trend from 1998 to 2014 using the TRMM data. Based on the trend from 1998 to 2014, trends for 2017, 2018, 2019, 2020, and 2021 were forecasted and compared against the mean flash densities calculated using the ISS dataset (2018 to 2021).

## 3. Data Analysis and Results

The spatiotemporal variation of lightning flash densities across Sri Lanka from 1998 to 2014 and 2018 to 2021 are presented in this section. The study was conducted by analyzing seasonal, annual, and monthly data obtained from the TRMM and ISS satellites.

### 3.1. Distribution of Lightning Flash Densities from 1998 to 2014

#### 3.1.1. Annual Variation

Average flash density from 1998 to 2014 varied between 6.00–14.00 flashes $km^{-2}$ $year^{-1}$ as shown in Table 1.

**Table 1.** All island annual zonal statistics from 1998 to 2014.

| Year | Unit (flashes $km^{-2}$ $year^{-1}$) | | | |
|------|---------|---------|---------|--------------------|
|      | Average | Maximum | Minimum | Standard Deviation |
| 1998 | 6.40    | 40.11   | 1.12    | 4.56  |
| 1999 | 7.00    | 36.72   | 1.12    | 4.67  |
| 2000 | 10.10   | 69.07   | 1.12    | 7.09  |
| 2001 | 6.73    | 45.59   | 1.12    | 5.19  |
| 2002 | 7.63    | 51.28   | 1.12    | 4.65  |
| 2003 | 8.17    | 75.86   | 1.12    | 6.85  |
| 2004 | 9.28    | 91.05   | 1.12    | 7.38  |
| 2005 | 11.50   | 54.03   | 1.12    | 7.90  |
| 2006 | 10.52   | 72.39   | 1.12    | 7.25  |
| 2007 | 9.23    | 56.12   | 1.12    | 7.06  |
| 2008 | 10.87   | 51.24   | 1.12    | 6.87  |
| 2009 | 8.82    | 54.20   | 1.12    | 5.60  |
| 2010 | 13.48   | 57.85   | 1.12    | 7.21  |
| 2011 | 13.89   | 65.49   | 1.12    | 8.84  |
| 2012 | 9.81    | 48.99   | 1.12    | 6.21  |
| 2013 | 8.81    | 61.18   | 1.12    | 6.35  |
| 2014 | 8.54    | 102.17  | 1.12    | 8.83  |

The two highest average flash densities were recorded in consecutive years 2011 and 2010, with 13.89 flashes $km^{-2}$ $year^{-1}$ and 13.48 flashes $km^{-2}$ $year^{-1}$, respectively. The lowest average flash density of 6.40 flashes $km^{-2}$ $year^{-1}$ was recorded in 1998. It recorded a maximum flash density of 102.17 flashes $km^{-2}$ $year^{-1}$ in 2014.

#### 3.1.2. Seasonal Variation

Average lightning flash densities from 1998 to 2014 for four seasons, First Inter-Monsoon (FIM), Southwest Monsoon (SWM), Second Inter-Monsoon (SIM), and Northeast Monsoon (NEM), are shown in Figure 3.

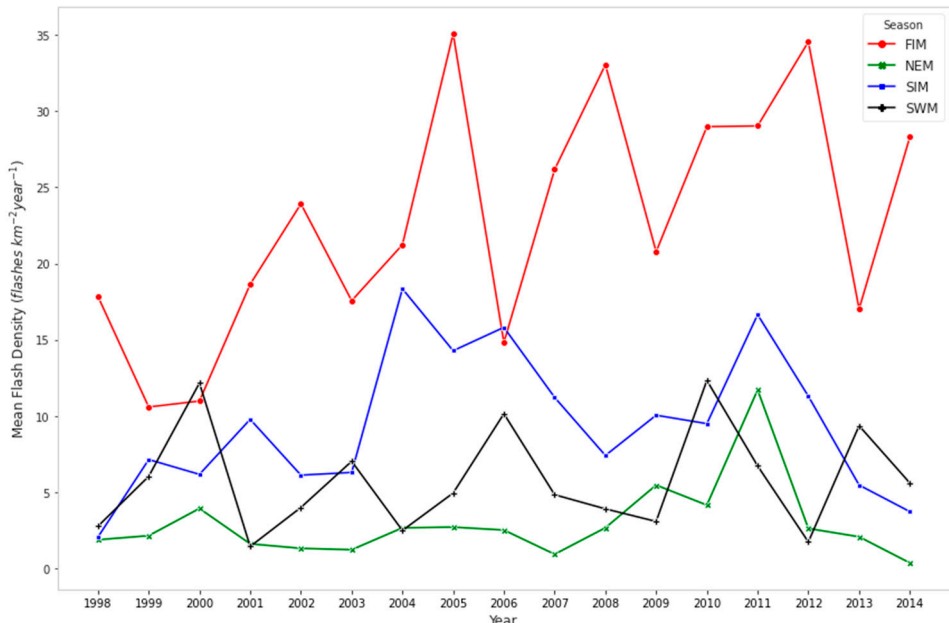

**Figure 3.** All island seasonal variation of mean lightning flash densities from 1998–2014.

FIM, which is in March and April, had recorded the highest number of flashes compared to other monsoon seasons. The highest flash density observed during the FIM was in 2005, with a value of 35.09 flashes km$^{-2}$ year$^{-1}$. Overall, the lowest flash densities were observed during the northeast monsoon season, with the lowest average flash density being 0.36 flashes km$^{-2}$ year$^{-1}$ in 2014. Compared to other seasons, FIM was most significant between 2006 and 2014, with values ranging between 15.00 and 35.00 flashes km$^{-2}$ year$^{-1}$. The peak of SIM occurred in 2004 and since then it has been gradually decreasing, with a sudden spike in 2011. SWM demonstrated a constant flow with a peak intensity that did not exceed 15.00 km$^{-2}$ year$^{-1}$. NEM, on the other hand, was low until 2007, then showed an increase until 2011, followed by a decline.

### 3.1.3. Monthly Variation

According to Figure 4, it can be clearly seen that over the period from 1998 to 2014, the highest number of flashes occurred during April, and that 2012 recorded the highest value of 60.45 flashes km$^{-2}$ year$^{-1}$ for this month, followed by 54.38 flashes km$^{-2}$ year$^{-1}$ in 2014. After April, the most frequent lightning occurred in October and March. Overall, July was the month with the lowest number of flashes. A considerable number of months with a lightning density of above 10.00 flashes km$^{-2}$ year$^{-1}$ occurred in the years 2000 and 2011. It is noteworthy that there was a significant level of lightning between February and May 2011. The mean flash densities in December were negligible during 2011, whereas in 2009 it was as high as 13.41 flashes km$^{-2}$ year$^{-1}$.

### 3.2. The Spatiotemporal Variation of Lightning Flash Densities from 2018 to 2021
### 3.2.1. Annual Variation—Country-Wise

Average lightning flash densities obtained from 2018 to 2021 are shown in Figure 5. The data for 2018 and 2019 displayed an increasing trend in lightning densities compared to 2020 and 2021, registering 8.21 flashes km$^{-2}$ year$^{-1}$ and 10.48 flashes km$^{-2}$ year$^{-1}$ respectively. The lowest value was observed in 2020 at 5.48 flashes km$^{-2}$ year$^{-1}$. The steep decrement from 2019 to 2020 may be because of the climatological changes that occurred due to the travel restrictions that were imposed during the COVID pandemic situation in the country.

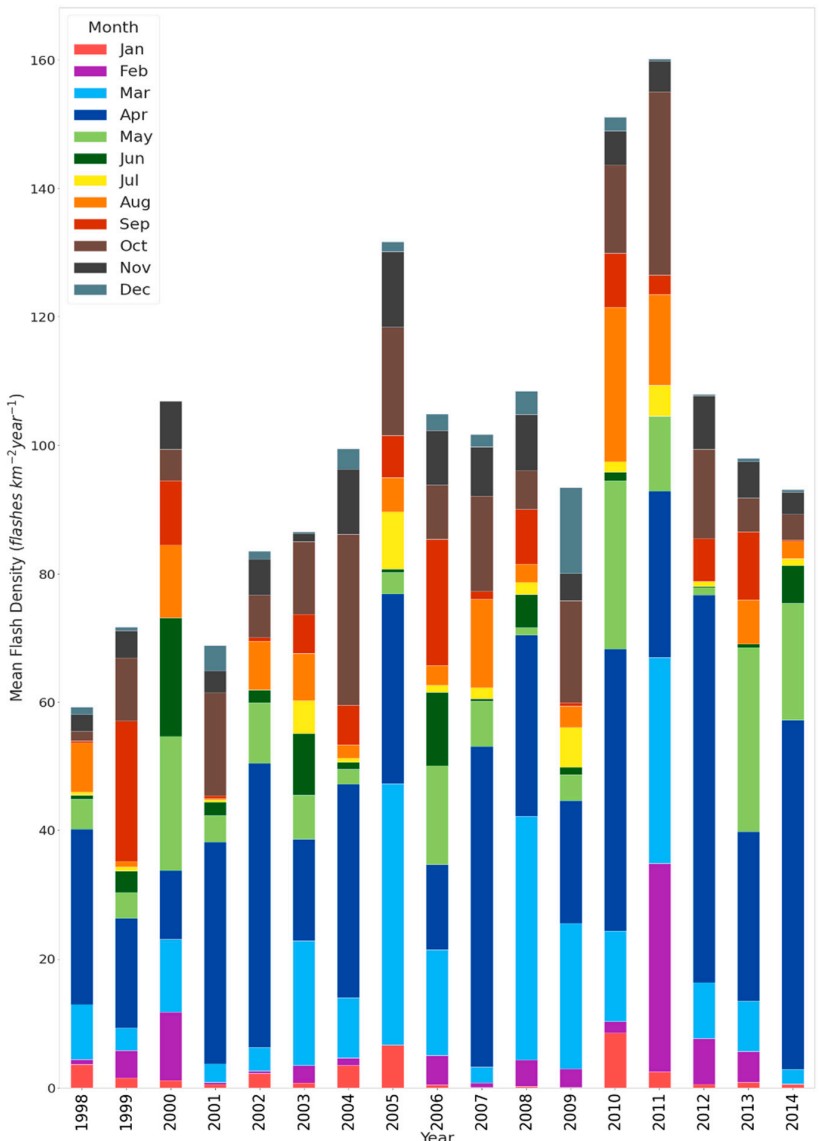

**Figure 4.** All island monthly lightning flash distribution from 1998 to 2014.

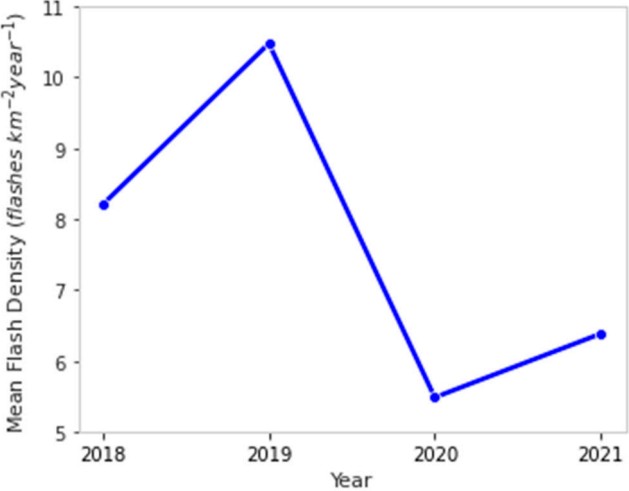

**Figure 5.** All island yearly variations from 2018–2021.

### 3.2.2. Annual Variation—District-Level

Overall, mean flash densities in 2019 in most of the districts displayed an increase compared to others, as shown in Figure 6. The highest mean flash density in 2019 was observed in Colombo at 34.85 flashes km$^{-2}$ year$^{-1}$, while Kegalle and Kurunegala displayed 27.42 flashes km$^{-2}$ year$^{-1}$ and 26.69 flashes km$^{-2}$ year$^{-1}$, respectively. Colombo, Kalutara, Kurunegala, Gampaha, Matale, and Matara showed a significant decrease in the mean flash densities in 2020. By a considerable margin, the year 2020 displayed the lowest average flash density across all districts. More interestingly, the average flash density in the Northern Province in 2018 was significant compared to the other three years. In the case of Ratnapura and Nuwara Eliya districts, their mean flash densities remained high from 2018 to 2020, but in 2021 it dropped to less than half. The aerosol content in the atmosphere directly affects lightning formation by enhancing the electrification process of the clouds [33]. A high aerosol content is available in an urban district like Colombo due to the high population and traffic conditions. This could result in obtaining the highest flash density in 2019, as depicted in Figure 6.

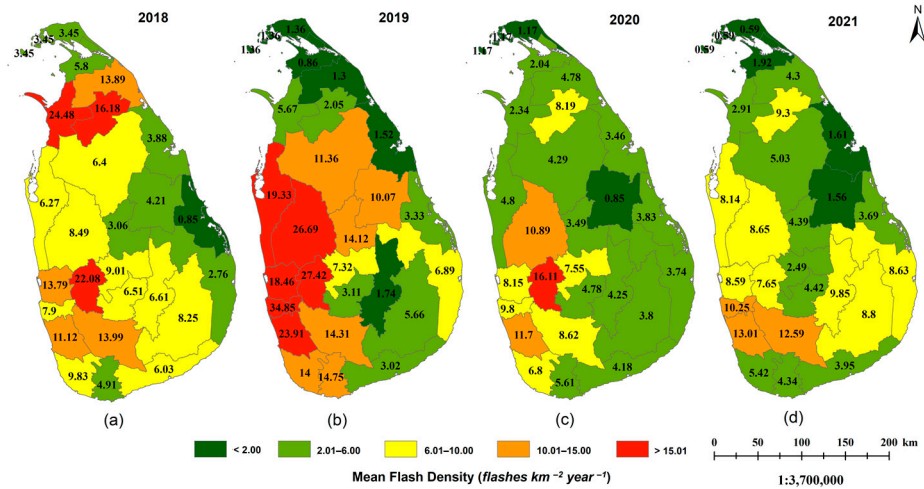

**Figure 6.** (**a**–**d**) represent district-level annual variation of lightning flash densities in 2018, 2019, 2020, and 2021 respectively.

### 3.2.3. Monthly Variation—Country-Level

April had recorded the highest flash densities in all four years, wherein the highest value of 50.47 flashes km$^{-2}$ year$^{-1}$ had been recorded in 2019. The second-highest peaks were visible in September, as shown in Figure 7. Flash density values during August, September, and October 2020 were significantly less compared to other years. While looking at the years 2018–2021, the year 2019 stood out as having the most lightning between February and June, while the years 2021 and 2018 were highlighted for having the highest strikes for the months of August-September and October-November, respectively. The overall behavior in 2020 were not significant over the months except for July and December which recorded the highest amount of lightning flashes compared to other years.

### 3.2.4. Monthly Variation—District-Wise

The mean flash density for each month, as shown in Figure 8, provides important information for each district. Overall, it is evident that April had the most lightning strikes over all the districts, whereas July displayed the least number of flashes. In comparison to the other three years, 2019 displayed the highest number of flash occurrences, while 2021 had the least. In 2018, the highest flash densities occurred in April for the Manner district, and in May for the Kegalle district. Colombo, Kegalle, Kurunegala, and Puttalam all experienced significant flashes in April 2018, and Colombo and Matara both experienced some noticeable flashes in July. Overall flash occurrences appear to be lower across districts

in 2020 and 2021, but Kegalle and Kurunegala experienced flashes in April 2020, and Rathnapura did so in April 2021.

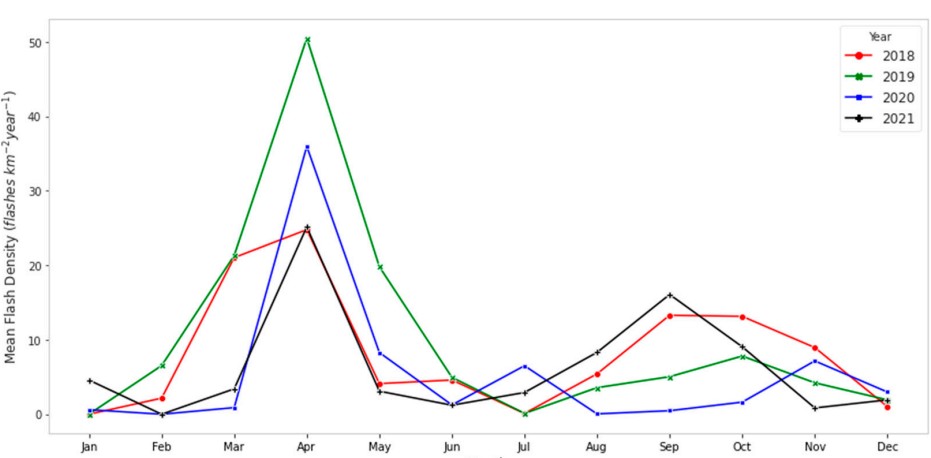

**Figure 7.** All island monthly variation from 2018–2021.

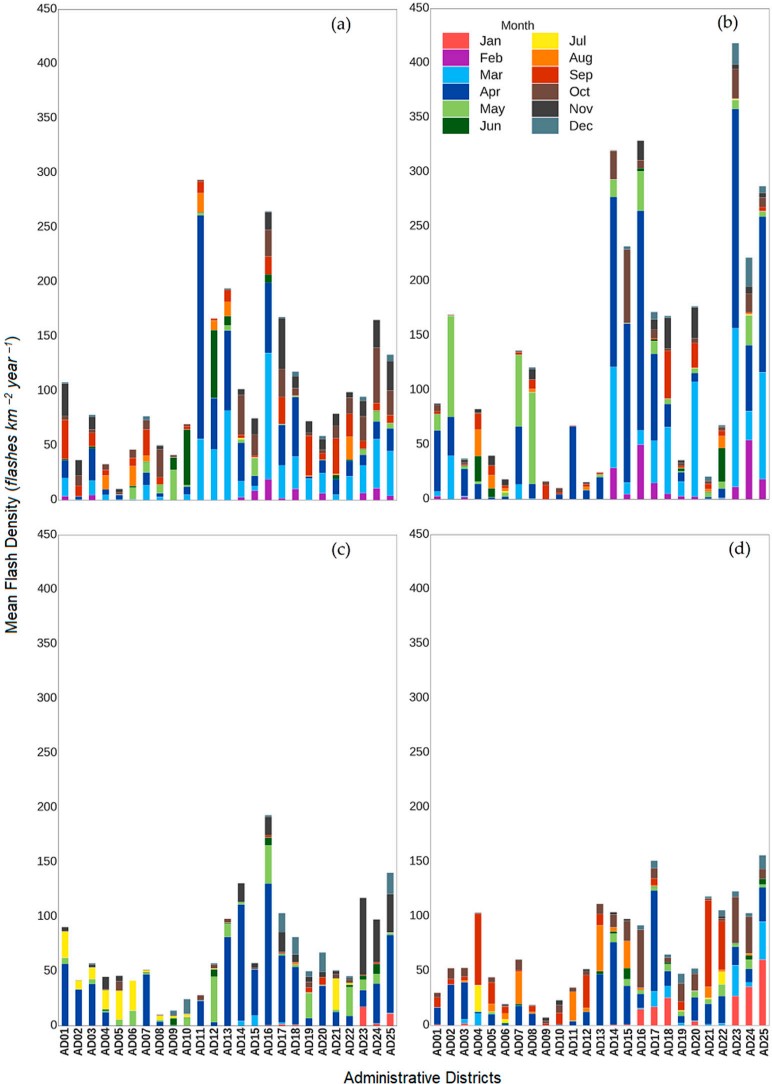

**Figure 8.** (**a**–**d**) represent the district-wise monthly variation of lightning flash densities from 2018 to 2021, respectively.

### 3.2.5. Seasonal Variation—Country-Wise

The highest flash density was recorded in 2019 during First Inter-Monsoon season, with a value of 35.89 flashes $km^{-2}$ $year^{-1}$. Overall, the least mean flash densities were recorded during the Northeast Monsoon season, and these values varied between 1.06 flashes $km^{-2}$ $year^{-1}$ and 2.82 flashes $km^{-2}$ $year^{-1}$. According to Figure 9, lightning flashes in 2020 during all four seasons were less than in 2018 and 2019.

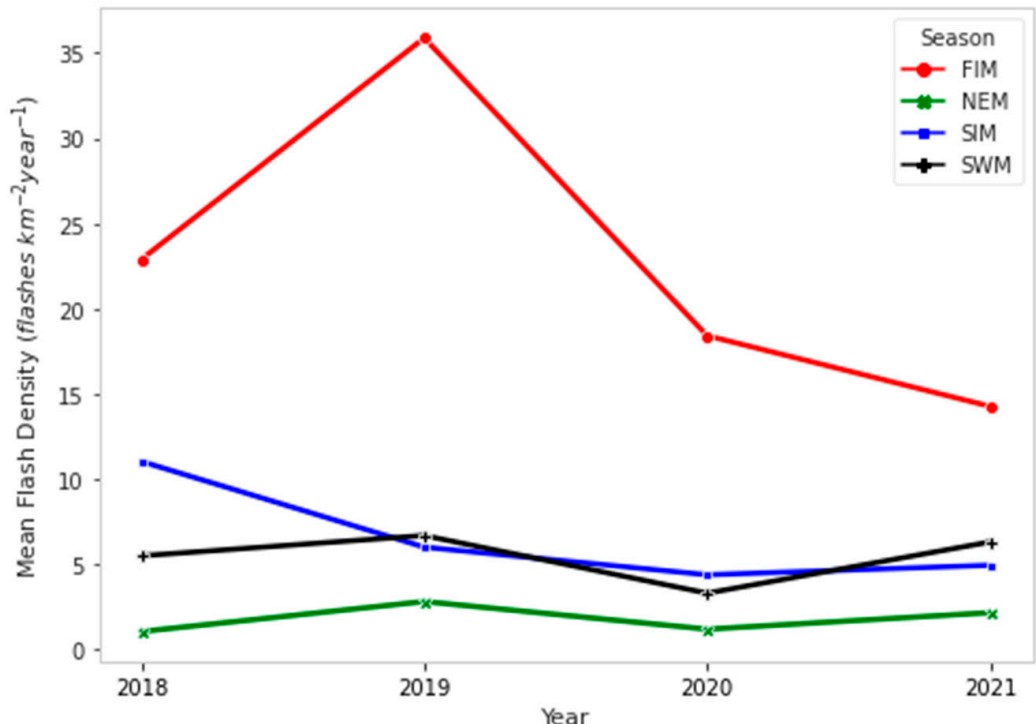

**Figure 9.** All island seasonal variations of light flash distribution from 2018–2021.

### 3.2.6. Seasonal Variation—District-Wise

Overall, lightning flash densities in all districts decreased from 2018 to 2021 according to Figure 10. FIM season indicated the most lightning strikes across the country, while the NEM season displayed the fewest. During the FIM in 2018 and 2019, clear boundaries were observed for flash densities above 130 flashes $km^{-2}$ $year^{-1}$ occurring in Manner in 2018 and in Colombo, Kalutara, Kurunegala, and Kegalle in 2019. Colombo and Rathnapura districts recorded 38.06 and 36.17 flashes $km^{-2}$ $year^{-1}$, respectively, during the 2018 SIM period, while only Puttalam and Colombo registered flash density values above 30 flashes $km^{-2}$ $year^{-1}$ in 2019 and 2020, respectively. The only two significant lightning strikes in any NEM season were represented in 2019 and 2018 for Gampaha and Kalutara districts, respectively, with values of 20 to 30 flashes $km^{-2}$ $year^{-1}$.

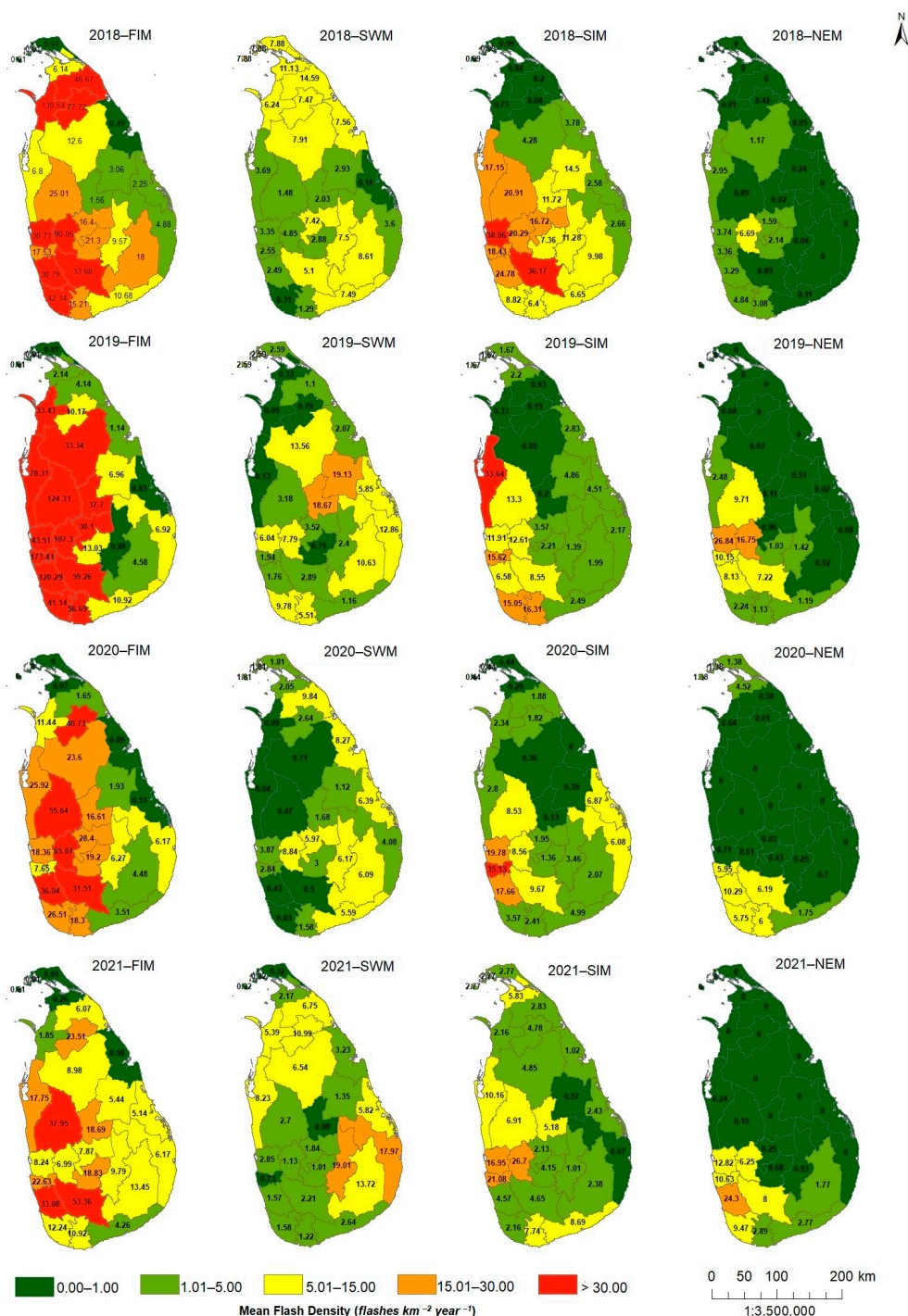

**Figure 10.** 2018–2021 monsoon season variation during FIM, SWM, SIM, and NEM.

## 4. Discussion

This section focuses mainly on discussing annual and seasonal trends in lightning using 17 years of historic lightning data in Sri Lanka. Analysis of these trends will help different parties such as farmers, construction workers, and government personnel to implement plans to reduce hazards that can be caused due to lightning.

Trend calculations were done using the Mann-Kendall test with 5% significant and 95% confidence levels. The Mann-Kendall test is a statistical test that is used to identify the existence of a trend based on the value obtained. Sen's slope estimation method was used to calculate the magnitude and the direction of the slope. Annual average lightning flash densities from 1998 to 2014 were used to calculate the trend, and the resulting statistical

parameters are shown in Table 2. The tau value of 0.35 and *S* statistic of 48.00 indicate an increasing trend which is statistically significant, as the *p*-value (0.05) obtained lies on the 0.05 significant level. The trend slope was 0.23, indicating a positive trend, and the annual lightning trend line was generated according to Equation (8). Figure 11 represents the trend line. Per Figure 11, it is visible that the mean flash densities varied over the duration from 1998 to 2014, with steep peaks and troughs, and this could have resulted in obtaining 0.23 as the trend slope. Even though there was an increase from 1998 to 2000, in 2001 flash density decreased to its second-lowest value. Again from 2001 to 2005, density increased gradually, whereupon it then fluctuated until 2009. A sudden increase and decrease were seen from 2009 to 2010, and from 2011 to 2012. Since 2012, it has decreased gradually.

**Table 2.** Mann-Kendall's and Sen's slope parameters for yearly calculations from 1998 to 2014.

| Statistic Measure | Value |
| --- | --- |
| *p* value | 0.05 |
| $Z_c$ | 1.94 |
| Tau | 0.35 |
| *S* | 48.00 |
| Slope | 0.23 |
| Intercept | 7.38 |

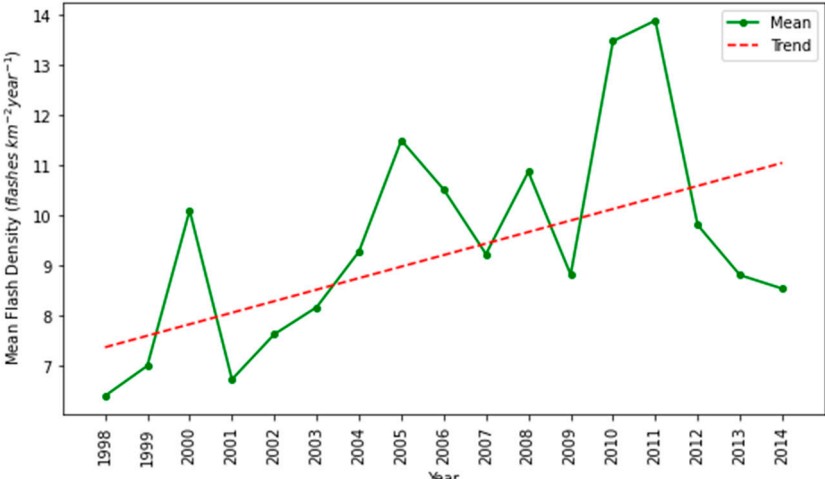

**Figure 11.** The annual trend line from 1998 to 2014.

The equation of the trend line is indicated in Equation (8).

$$y = 0.23 \times (\text{year}) + 7.38 \tag{8}$$

It is a challenge to identify patterns and trends in climatological datasets. Per Table 2, it is visible that the trending pattern considering yearly lightning data from 1998 to 2014 is small. Annual lightning flash densities for the years from 2015 to 2021 are forecasted by creating the trend line using Equation (8) and represented in Figure 12. Data obtained from the ISS from 2018 to 2021 are also plotted in the same figure. It is observed that there is a noticeable difference between the forecasted and actual values from 2018 to 2021. Edirisinghe, M, et al. recorded the average flash density (mean) from 1998 to 2014 as 8.26 flashes $\text{km}^{-2}$ $\text{year}^{-1}$ and the standard deviation (STD) as 3.88 flashes $\text{km}^{-2}$ $\text{year}^{-1}$ [13]. Forecasted flash densities for 2018 to 2021 lie within the range of mean $\pm$ STD, which validates the forecasted values.

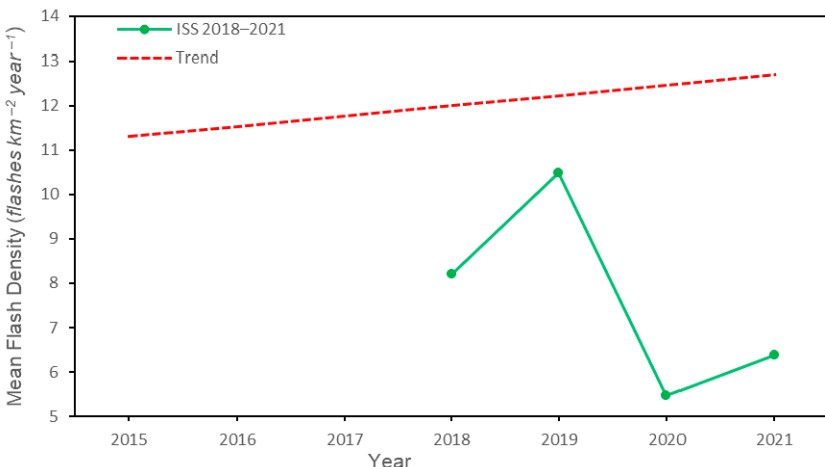

**Figure 12.** Yearly forecasted values from 2015 to 2021.

The Mann-Kendall test and Sen's slope estimator were applied to seasonal data from 1998 to 2014 considering the First Inter-Monsoon (FIM), Southwest Monsoon (SWM), Second Inter-Monsoon (SIM), and Northeast Monsoon (NEM). The results obtained along with Sen's slope values are represented in Table 3.

**Table 3.** Mann-Kendall test and Sen's slope parameters for seasonal calculation from 1998 to 2014.

| Season | $p$ Value | $Z_c$ | Tau | $S$ | Slope | Intercept |
|--------|-----------|-------|------|-------|-------|-----------|
| FIM | 0.03 | 2.18 | 0.40 | 54.00 | 0.96 | 13.59 |
| SWM | 0.65 | 0.45 | 0.09 | 12.00 | 0.11 | 4.05 |
| SIM | 0.54 | 0.62 | 0.12 | 16.00 | 0.17 | 8.11 |
| NEM | 0.65 | 0.45 | 0.09 | 12.00 | 0.04 | 2.17 |

In the case of FIM, a positive Tau value (0.40) and a positive $S$ statistic (54.00) showed an increasing trend and the $p$-value $< 0.05$ revealed that the trend was statistically significant. On the other hand, even though SWM, SIM, and NEM had positive tau and $S$ statistics indicating an increasing trend, all $p$ values were greater than 0.05, indicating that trends were not statistically significant over the period. The slope of the FIM obtained through Sen's slope was 0.96 and the intercept was 13.59. The trend values of SWM and SIM were 0.11 and 0.17 respectively. The trend with the least slope was observed during NEM with a value of 0.04. Equation (9) represents the equation used to generate the trend line of the first inter-monsoon.

$$y = 0.97 \times (\text{year}) + 13.59 \tag{9}$$

Figure 13 denotes the trend of FIM from 1998 to 2014, which is calculated using Equation (9). Even though there was a drop in 1999, flash densities increased gradually until 2002. After the increase in 2005, there was a sudden drop in 2006, but the trend continued to rise until 2008. After the sudden drop in 2009, values increased gradually until 2012. The second-largest decrease occurred in 2013, followed by an increase in 2014. Figure 14 represents the forecasted lightning flash densities for the first inter-monsoon seasons from 2015 to 2021. Data relating to FIM collected from the ISS for 2018 to 2021 is also depicted in the same graph. As identified by [13], for lightning occurrences from 1998 to 2014, the average flash density during FIM was 22.25 flashes km$^{-2}$ year$^{-1}$ and the standard deviation was calculated at 15.09 flashes km$^{-2}$ year$^{-1}$. Forecasted values for the FIM for the period from 2018 to 2021 lie within the range of mean $\pm$ STD which provides validation for the forecasted values. However, the forecasted value in 2019 tends to align with the actual values, whereas there is a clear difference in actual and forecasted values in 2020 and 2021, which could have resulted due to environmental changes that happened during the lockdown period. Overall, the increasing trend in the FIM could be

used in future studies to forecast the flash densities, as well as to fill the missing data in past recordings that occurred due to the technical unavailability of the sources.

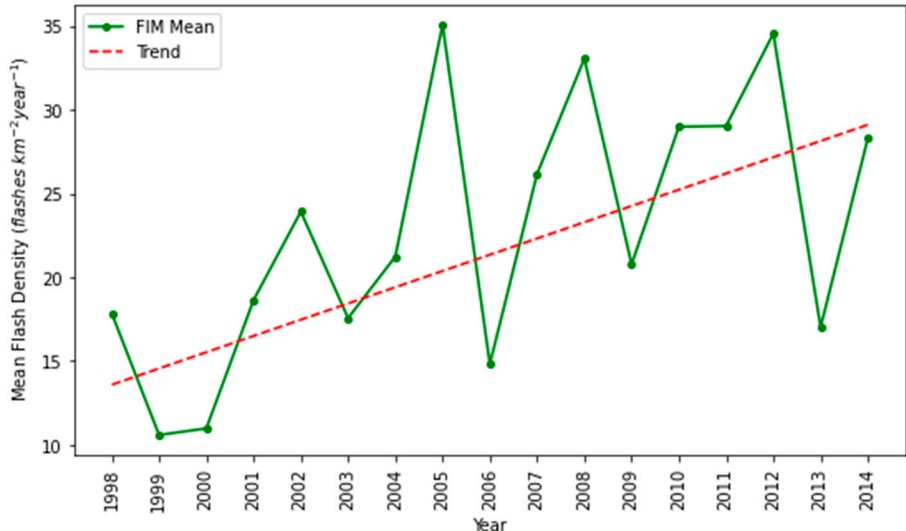

**Figure 13.** The trend line for FIM from 1998 to 2014.

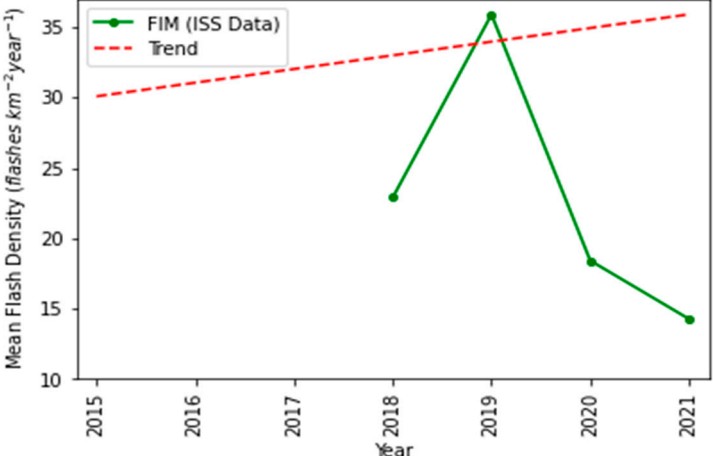

**Figure 14.** Lightning flash densities forecasted for FIM from 2015 to 2021.

Identifying trends in lightning during FIM, SWM, SIM, and NEM considering administrative districts will reduce the risk of lightning strikes during farming and construction. In order to identify existing seasonal trends in the administrative districts in Sri Lanka, the Mann-Kendall test and Sen's slope estimator were performed on seasonal data from 1998 to 2014. Sen's slopes obtained for each district are displayed using colour gradients and tau values are displayed as numbers, as shown in Figure 15. The highest and second-highest positive trends during FIM are shown in Kalutara and Galle districts with slope values of 3.27 and 1.7, respectively, whereas negative trends are observed in Kandy, Matale, and Polonnaruwa districts. During SWM, Vavuniya and Mullaitivu show positive trend values of 0.45 and 0.37, respectively. Districts such as Ampara, Badulla, and Moneragala display the highest negative trends during SWM. Ratnapura district shows the highest trend of 0.61 during SIM, whereas the highest negative trend is displayed by Gampaha, with a −0.54 trend value. Apart from Gampaha, which displays a positive trend of 0.32, and Galle, with a −0.58 trend, all other districts display their lowest trend values during NEM. By obtaining statistical parameters using the Mann-Kendall test for NEM, it can be concluded that no district shows a statistically significant trend in lightning.

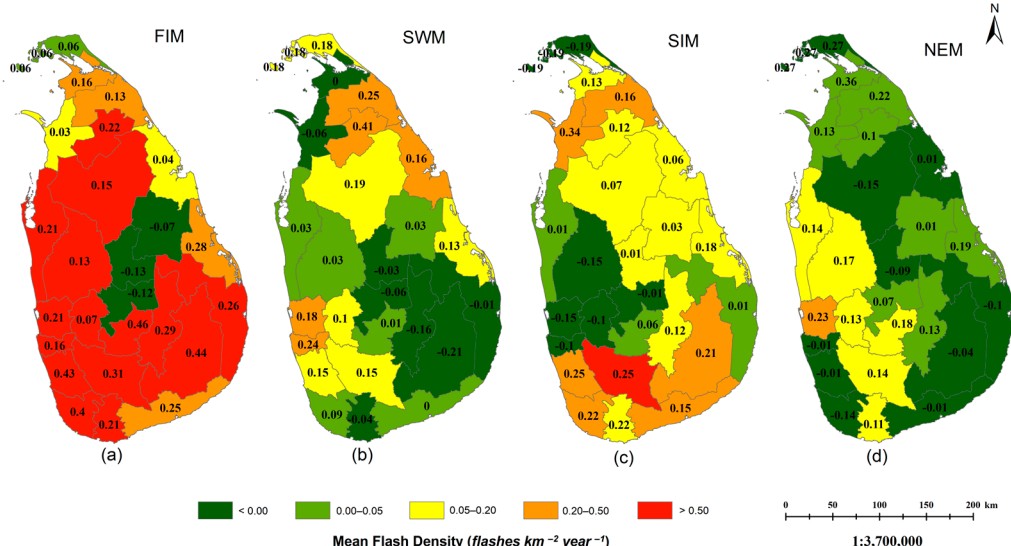

**Figure 15.** District-level Sen's slope (colour gradient) and Kendall's Tau values (numbers) for four seasons, FIM, SWM, SIM and NEM, represented by (**a**–**d**), respectively.

Seventeen years of data collected using the TRMM were used for the annual and seasonal trend calculations. After the discontinuation of LIS on the TRMM, LIS on the ISS was sent in 2017. Due to the missing lightning data from 2015 to 2017 and the dataset not being sufficiently large, datasets from 2018 to 2021 were not considered in the trend calculations. The study done by Yusfiandika and others identified that the air temperature and humidity were greatly changed during the lockdown period in 2020 and caused the lightning flash count in Europe and Oceania to drop significantly [30]. As visible in Figures 12 and 14, the sudden drop in lightning flash densities in 2020 might have occurred due to the changes in these environmental factors during the lockdown period in Sri Lanka in 2020. However, further investigations need to be conducted to identify the variation of climatic parameters that have a direct impact on lightning behavior, such as humidity and air temperature, during the lockdown period over Sri Lanka in 2020. Future investigations could be used to understand the variation of lightning that is visible in Figures 12 and 14.

The methodology followed in this study could be applied to identify lightning variation in tropical and subtropical regions. This would be beneficial in identifying the lightning distribution over the past few years and also to forecast flash densities in Sri Lanka. The results of this study can be used as an indicator to identify lightning patterns in Sri Lanka and detect districts that are more prone to lightning, and take precautions to mitigate lightning hazards by employing proper lightning protection systems.

For this study, total lightning data (cloud-to-cloud, cloud-to-air, cloud-to-ground, and ground-to-cloud) collected by the LIS on the TRMM from 1998 to 2014 and the LIS on the ISS from 2018 to 2021 were used. Due to the limitation of observational times of the satellites over the country, there is a possibility that the overall lightning activities recorded by the satellites might differ slightly from the real-time occurrences, which is a major limitation of this study. After the termination of the TRMM in April 2015, a missing data period from 2015 to 2017 existed until the launch of the LIS on the ISS in February 2017. The existence of this gap period and the availability of ISS data for a short period of time (2018–2021) may have an impact on the trend analysis and the forecasting. Lightning Location System (LLS) is one of the proper ways to monitor lightning activity over a region with greater accuracy compared to satellites. However, due to the fact that Sri Lanka is distinctly lacking in its provision of a ground-based lightning location system, the optimum solution is to use satellite lightning data to explore the lightning distribution and trends across the country. Even taking into account these limitations, the proposed approach can be successfully applied across the globe where satellite lightning observations are available.

## 5. Conclusions

The study was conducted to evaluate the spatial and temporal variability of lightning over Sri Lanka from 1998 to 2014 using TRMM data, and 2018 to 2021 using ISS data. The Mann-Kendall Test and Sen's slope, two of the most effective nonparametric methods for identifying monotonic trends in climatological data, were used to achieve this result.

Initially, datasets from 2018 to 2021 were used to generate district-level and island-level lightning density on an annual, seasonal, and monthly basis. Based on the data, the number of flashes across the whole island changes every year; 2019 was found to have the highest peak, while 2020 had the lowest. According to 2019's annual data broken down by district, the highest flash density was observed in the Colombo district. Additionally, districts on the west and northwest sides of the island showed distinct margins for high flash densities in 2018 and 2019, whereas no such margins emerged in 2020 or 2021. According to the monthly variation, April depicted the highest value from 2018 to 2021 annually. The monthly data from March to May and from August to November depicted the features of two distinct humps, and February to June represented the highest hump. District-wise monthly variation indicated that most of the districts were stroked by lightning in April 2019. The results showed that April was the month with the highest number of lightning strikes across the districts and the numbers had slightly decreased in 2020 and 2021. FIM season depicted a significant amount of flash density compared to the other seasons when considering the whole country, and its peak was reached in 2019. Even though FIM demonstrated more flashes on the northern side of Sri Lanka in 2019, most of the districts in the wet zone and intermediate zone were impacted in 2018, 2020, and 2021. The other significant season was SIM, where district-wise, the impact took place around the wet zone, and it was found to be significantly lower compared to the FIM.

Annual and seasonal trend calculations were performed using the TRMM data from 1998 to 2014 by applying the Mann-Kendall test and Sen's slope estimator. Considering annual lightning data, it showed that the trend slope was as low as 0.23, due to the high fluctuation of the data. The seasonal data from 1998–2014 was then subjected to the Mann-Kendall test and Sen's slope estimator, which revealed a positive trend for the first inter-monsoon that could be applied to future forecasting. According to the results, a strong positive trend exists during the FIM with a slope value of 0.956, while the remaining seasons did not exhibit a significant trend. Additionally, district-wise seasonal data was added to the test, and a significant slope was found in the Kaluthara district during the FIM seasons from 1998 to 2014. More importantly, the results showed significant levels of flash density across districts for the FIM season. Therefore, the observed FIM trend results can be utilized to forecast the future lightning flash densities during FIM seasons in Sri Lanka.

**Author Contributions:** Conceptualization, Vindhya Kalapuge, Dilaj Maduranga, Niranga Alahacoon and Mahesh Edirisinghe; Data curation, Vindhya Kalapuge; Formal analysis, Vindhya Kalapuge and Dilaj Maduranga; Investigation, Vindhya Kalapuge and Dilaj Maduranga; Methodology, Vindhya Kalapuge, Dilaj Maduranga, Niranga Alahacoon and Mahesh Edirisinghe; Resources, Mahesh Edirisinghe; Supervision, Mahesh Edirisinghe and Rushan Abeygunawardana; Validation, Niranga Alahacoon, Mahesh Edirisinghe and Rushan Abeygunawardana; Visualization, Vindhya Kalapuge, Niranga Alahacoon and Mahesh Edirisinghe; Writing—original draft, Vindhya Kalapuge; Writing—review and editing, Dilaj Maduranga, Niranga Alahacoon, Mahesh Edirisinghe and Manjula Ranagalage. All authors have read and agreed to the published version of the manuscript.

**Funding:** This research received no external funding.

**Institutional Review Board Statement:** Not applicable.

**Informed Consent Statement:** Not applicable.

**Data Availability Statement:** Lightning flash data are available at Lightning Imaging Sensor (LIS) on the Tropical Rainfall Measurement Mission (TRMM) and International Space Station (ISS) of NASA (https://lightning.nsstc.nasa.gov/data/data_lis.html [accessed on 21 June 2022]).

**Acknowledgments:** The Global Hydrology Resource Centre (GHRC), Marshall Space Flight Centre (MSFC), Huntsville, USA is gratefully acknowledged for providing the LIS Science data. Assistance and facilities provided by the Department of Physics, University of Colombo are similarly gratefully acknowledged.

**Conflicts of Interest:** The authors declare no conflict of interest.

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
