# Peer review of "Overview of Lightning Trend and Recent Lightning Variability over Sri Lanka"

_ijgi, doi:10.3390/ijgi12020067_

Round 1
Reviewer 1 Report
Review of the manuscript “Overview of Lightning Trend and Recent Lightning Variability 2 over Sri Lanka” by Vindhya Kalapuge et al.
General comments.
The manuscript analyses the lightning activity over Sri Lanka using lightning data provided by LIS on board of the Tropical Rainfall Measuring Mission (TRMM), during the period between 1998 and 2014, and data of LIS instrument placed on the International Space Station (ISS), during the period between 2018 and 2021. The analysis was performed annually, seasonally and monthly, on district-wise and island-wise. By using the Mann-Kendall test and the Theil-Sen test, the authors identified the trend on lightning activity over the island.
The analysis shows a positive trend on the annual lightning activity and a strong positive trend during the first inter-monsoon while the remaining seasons didn’t show a significant trend. The results found can be used to forecast lightning activity in Sri Lanka to develop comprehensive protective for building and human. However, the trend found using lightning data for the period between 1998 and 2014 does not forecast adequately the lightning activity detected for the period between 2018 and 2021. The noticeable difference, between the forecast and the detected values, detracts the trend analysis performed.
The manuscript is well written and the presentation is well structured and clear. Most of the figures are adequate and support the analysis and results presented on the study.
However, the authors should discuss some issues before the manuscript is ready for publication.
Major comments.
1. My major concern is about the calculation of the flash density given by equations (6) and (7). On these equation, a summation of flash detected by LIS is performed, but it is not clear over which period of time is the summation done (daily, monthly or annually?). The summation is multiplied by 24 (hour?) x 3600 (seconds?), which suggest that the summation of flash is performed daily. If so, to compute the flash density per year is necessary also to multiply this quantity by 365 days. Besides, why the summation of flash is divided by twice the viewtime of LIS (90 or 80 seconds)?
On Lines 143 and 151, the authors indicated that LIS, on board of TRMM, has an overhead time of 80 seconds while LIS, on board of ISS, a 90 s window period for observations (values which are used on eq 6 and 7). However, this time windows are for ideal cases. According to Christian (2000), the viewtime of LIS on different locations can have a wide range, between 0 to 80 (90) seconds. The real viewtime can be find on LIS file data, which is reported as Effective_obs. This parameter represents the total viewing time for each location along the LIS field of view and should be taken into account for the flash density calculation.
Therefore, I believe that the flash density computed and used to perform the analysis on the manuscript is not well determinate and needs to be corrected.
Christian, H. J. (2000). Algorithm theoretical basis document (ATBD) for the Lightning Imaging Sensor (LIS). http://thunder. msfc. nasa. gov/bookshelf/pubs/atbd/.
2. Figure 3 (seasonal variation) and 4 (monthly variation) seem to show the same information. The authors found that April recorded the highest number of flashes, followed by October and March. And they also report that the First Inter-Monsoon (FIM), which includes April and March, recorded the highest number of flashes, followed by the Second Inter-Monsoon (SIM), which include October and November. It seems to me, that figure 4 does not add new information to the manuscript.
3. Lines 271-274. The authors associate the decrement on the lightning activity from 2019 to 2020 to climatological changes due to the restrictions impose during the COVID pandemic in the country. Is this purely speculative or there are indications that some parameters associated with lightning production (as for example air pollution) had changed on this period of time?
4. In figure 6 and 10, it is showed the district-level annual and seasonal variation of lightning flash densities, respectively. Why it was chosen a district-level spatial resolution to analyze the spatial distribution of lightning activity? Given that lightning activity “varies due to longitude, latitude, altitude, soil content, topography, and, the vegetation of the region” (lines 72-73) and the description of the topography and the distinctions of different zones, according to the annual rainfall (lines 110-119), over the island, it seems no the best way to present and discuss the spatial resolution of lightning distribution. Please clarified.
5. Figure 8 is hard to read and does not seem to add new and different information that it was presented in figure 6, 7 and 9 (Please, see comment 2)
6. The authors found a linear trend for the annual lightning activity and the lightning activity during the First Inter-Monsoon season using data for the period between 1998 and 2014. Using these trend, the authors intend to forecast the lightning activity. However, the lightning activity detected for the period between 2018 and 2021 is not well forecast by the trend. These difference need to be discuss by the authors.
Minor corrections:
1. Lines 76. “… that of temporal regions in the …”. I believe the authors meant to refer “temperate regions”. If not please clarified
2. Line 115-118. Why it is used the quotations marks on Wet, Dry, Arid and Intermediate Zone?
3. Lines 173. Eq. (1). On the first summation on the definition of S, the upper limit should be instead of .
4. Lines 182. Eq. (2). I understand that the term with the summation should be instead of .
5. Line 258. “…In the controversy, July was …” I don´t understand the controversy.

Author Response
Author's Reply to the Review Report (Reviewer 1) is attached.

Reviewer 2 Report
This paper analyzed spatial and temporal variation of lightning activities in Sri Lanka by using the data of TRMM LIS and ISS. As authors pointed out in the paper, to understand lightning activities in Sri Lanka is important to taking measures to mitigate damage caused by lightning. I found several minor points to fix in your paper. I suggest you to revise your paper with reference to the following comments.
Minor comments:
1. L134 divisons → divisions
2. L.181-182, L.186-187, L.212-213 I recommend you to write each equation in a line.
3. L. 281-283 Not everyone is familiar with administrative divisions of Sri Lanka such as Colombo, Kalutara, Kurunegala and so on. To add the name of administrative divisions into Figure 1 is helpful for readers.
4. L.296 Figure 8 → Figure 7
5. Figure8 Characteristics in the figure is too small to read.
Author Response
Author's Reply to the Review Report (Reviewer 2) is attached.

Reviewer 3 Report
Overview:
This is a useful and reasonably well-written paper that examines the spatio-temporal variability of lightning over Sri Lanka in the period 1998 – 2014 (TRMM/LIS observations) and 2018 – 2021 (ISS/LIS observations). State-of-the-art statistical methodologies are applied to tease out lightning trends. The results of the study reasonably demonstrate that the methodology can help to identify lightning patterns in Sri Lanka and detect districts that are more prone to lightning (and that therefore are in need of better mitigation of lightning-caused hazards and lightning protection systems). This is particularly useful information (even if not perfect) given that Sri Lanka has a distinct lack of ground-based lightning detection systems. The paper is quite thorough in providing the annual, seasonal, and monthly variability of lightning. The figures and tables are very helpful and have been well-organized. The authors exhibit great care in applying the statistical methodologies, and I believe other investigators will find that their approach can be successfully applied elsewhere across the globe where satellite lightning observations are available. So one should note that the beneficial impact of this paper is that the methodology employed is useful far beyond just Sri Lanka.
The only concern I have, and it is likely a relatively minor one, is that the TRMM orbit altitude was boosted in August 2001 from 350 km to 402.5 km, and this would affect the viewtime value employed in the author’s Equation (7). See specific commentary to follow regarding Line 213 & 214. It is for this reason that I recommend publication after minor revisions.
Specific Required Revisions:
Line 16 and 24 of Abstract: change “lightning activities” to “lightning activity”. Make similar change throughout manuscript using an electronic scan of the given phrase.
Line 23 of Abstract: introduce the acronym IFM here, not later on line 29.
Line 63: Change “Kenneth” (the lead authors first name) to “Cummins” (his last name).
Line 68: The phrase “variation trend” is somewhat awkward; I would change it simply to “trend”. Do similar elsewhere in manuscript. Alternatively, you could say “… to analyze the variation in the trend of lightning activity over Sri Lanka.”
Line 73: It would be a good idea to add a sentence or two explaining why lightning activity depends on soil content and vegetation, as these two are not normally indicated, and the effect of these two are likely indirect effects and not as important as the other items that you list.
Line 98: Change “lighting” to “lightning”. … you dropped the “n”.
Line 104: Change “guild lines” to one compound word “guidelines”. … note your typo “d” (in “guild”) instead of “e”.
Line 111: “10.00o degrees” is redundant on the physical label. It should just be “10.00o”. Delete the space between the number and degree label “o”.
Line 143: TRMM/LIS nadir resolution is closer to 4km (not 3 km).
Line 144: It also measures the flash area (i.e., flash optical illumination area at cloud-top). It of course also records the time of the flash. Technically, it does not directly measure the flash energy (as does GLM), but rather the flash energy density (i.e., time integrated radiance) … to clarify this you could reference the Appendix of Koshak 2010:
· Koshak, W. J., 2010: Optical characteristics of OTD flashes and the implications for flash-type discrimination, J. Atmos. Oceanic Technol., 27, 1822-1838.
Line 145: The 350 km orbital altitude is only for the pre-boost period. In August 2001, the TRMM/LIS orbit altitude was boosted up to 402.5 km.
Line 150: The orbital altitude of ISS LIS is about 420 km (and varies a little), so is not the large value of 550 km you seem to indicate.
Line 152: What do you mean by “Novel”? … replace it with “ISS”. The ISS LIS does not have a 90% flash detection efficiency, but rather is comparable or slightly lower than TRMM/LIS’s flash detection efficiency (that you cite earlier). In the Blakeslee et al., 2020 paper they state: “Initial comparisons with those other sensors suggest [ISS LIS] flash detection efficiency around 60% (diurnal variability of 51–75%) …”
Line 153: Change “lighting” to “lightning”.
Line 213 & 214:
· You are rounding off (for the 0.2 deg x 0.2 deg specification) the grid area to 22 km x 22 km (at least as shown in these equations). You might indicate that you carry more significant digits in the computation if that is indeed true.
· You appear to be properly correcting for view-time by dividing sec/day by viewtime/day. However, because of the orbital boost of TRMM in Aug 2001 (as indicated above), the viewtime would change but no such change is accounted for in the equation (7) shown. Not sure what impact the change has on your results, but it should be examined. That is, the viewtime is larger at the higher orbit altitude of 402.5 km starting in Aug 2001 (compared to the 350 km pre-boost altitude period).
· However, you need to clarify to the reader (in several sentences) exactly how you came up with the numbers 2 x 90 sec/day (for ISS LIS) and 2 x 80 sec/day (for TRMM/LIS). For example, in the case of TRMM/LIS, I looked at a poster presentation that showed viewtime of 105 hrs/13 yrs which is equivalent to about 80 sec/day, or a factor of a half of what you get in your 2 x 80 sec/day value. So perhaps you need to explain the factor of 2 better. It might have something to do with the implicit grid areas used in the viewtime numbers, but not sure. So clarification is needed.
Line 269: Change “lighting” to “lightning”.
Line 274: You would want to explain & reference this a bit by noting the “aerosol effect” wherein air pollutants lead to enhancements in lightning as discussed in Williams (2005):
· Williams, E. R., Lightning and climate: A review, Atmo. Res., 76, 272-287, 2005.
Line 323 (Figure 8): It is difficult to read the label text on the vertical plot axis.
Line 328: Change “lighting” to “lightning”.
Author Response
Author's Reply to the Review Report (Reviewer 3) is attached.

Reviewer 4 Report
This study is important for monitoring lightning activities and for developing the related protective measures.
There are some points:
1) In section 2.2, the statement that TRMM "has an altitude of 350 km" is not accurate. The TRMM satellite was boosted from an average altitude of about 350 km before August 2001 to about 400 km after August 2001. Meanwhile, the TRMM LIS observes lightning for ~90 s as it passes overhead. Thus, is the 80 s in Equation (7) accurate? Suggest a clearer description of the calculation of time in Equation (7)?
2) The format of some equations is messed up and needs to be adjusted, such as Equation (2), (3), and (6). In addition, please pay attention to the normalized use of orthographic and italic type in equations.
3) Some writing styles should be uniform, such as Equation or Eq. in lines 208 and 209.
4) Why the analysis from 1998 to 2014 focuses more on interannual and seasonal variation, while the analysis from 2018 to 2021 focuses more on regional variation?
5) Figure 8 cannot be seen clearly. The lightning density in different regions is suggested to be represented by a spatial distribution figure.
6) The predicted values during the FIM are more consistent with the actual values than in other seasons. Have the reasons for this been considered?
Author Response
Author's Reply to the Review Report (Reviewer 4) is attached.

Round 2
Reviewer 1 Report
2th Review of the manuscript “Overview of Lightning Trend and Recent Lightning Variability over Sri Lanka” by Vindhya Kalapuge et al.
This is my second revision of the manuscript and I believe that the authors have adequately addressed all my previous concerns. Therefore, I consider the study is ready for publication.
Author Response
We sincerely thank you and highly appreciate your time and effort in reviewing our manuscript. Your valuable remarks, constructive feedback with helpful comments, suggestions, and careful corrections have guided us to enhance the overall quality of the revised manuscript.